# Hydrogen Peroxide Inhibits Hepatitis C Virus Replication by Downregulating Hepatitis C Virus Core Levels through E6-Associated Protein-Mediated Proteasomal Degradation

**DOI:** 10.3390/cells13010062

**Published:** 2023-12-28

**Authors:** Hyunyoung Yoon, Kyung Lib Jang

**Affiliations:** 1Department of Integrated Biological Science, The Graduate School, Pusan National University, Busan 46241, Republic of Korea; yhypia@pusan.ac.kr; 2Department of Microbiology, College of Natural Science, Pusan National University, Busan 46241, Republic of Korea; 3Microbiological Resource Research Institute, Pusan National University, Busan 46241, Republic of Korea

**Keywords:** HCV Core, hepatitis C virus, hydrogen peroxide, proteasome, E6-associated protein, p53

## Abstract

Hepatitis C virus (HCV) is constantly exposed to considerable oxidative stress, characterized by elevated levels of reactive oxygen species, including hydrogen peroxide (H_2_O_2_), during acute and chronic infection in the hepatocytes of patients. However, the effect of oxidative stress on HCV replication is largely unknown. In the present study, we demonstrated that H_2_O_2_ downregulated HCV Core levels to inhibit HCV replication. For this purpose, H_2_O_2_ upregulated p53 levels, resulting in the downregulation of both the protein and enzyme activity levels of DNA methyltransferase 1 (DNMT1), DNMT3a, and DNMT3b, and activated the expression of *E6-associated protein (E6AP)* through promoter hypomethylation in the presence of HCV Core. E6AP, an E3 ligase, induced the ubiquitin-dependent proteasomal degradation of HCV Core in a p53-dependent manner. The inhibitory effect of H_2_O_2_ on HCV replication was almost completely nullified either by treatment with a representative antioxidant, N-acetyl-L-cysteine, or by knockdown of *p53* or *E6AP* using a specific short hairpin RNA, confirming the roles of p53 and E6AP in the inhibition of HCV replication by H_2_O_2_. This study provides insights into the mechanisms that regulate HCV replication under conditions of oxidative stress in patients.

## 1. Introduction

An estimated 2–3% of the world population suffers from chronic hepatitis C virus (HCV) infection [1]. HCV significantly contributes to various hepatic diseases in humans, including hepatitis, liver cirrhosis, and hepatocellular carcinoma (HCC) [2,3]. As a member of the *Flaviviridae* family, HCV possesses a positive-stranded RNA genome of 9.6 kb, encoding a large polyprotein of approximately 3000 amino acids. This HCV polyprotein undergoes cleavage by both cellular and viral proteases, leading to the generation of three structural proteins (Core, E1, and E2) and seven non-structural proteins (NS1, NS2, NS3, NS4A, NS4B, NS5A, and NS5B) [4]. Among these viral proteins, Core has attracted special attention due to its multifunctional activities implicated in both viral replication and pathogenesis. For instance, HCV Core not only affects various intracellular signaling pathways in the cytoplasm but also regulates cellular gene expression in the nucleus, both of which are implicated in HCV-mediated HCC formation [5,6]. Additionally, HCV Core plays a pivotal role in determining the HCV replication rate, primarily due to its crucial function as a component of the nucleocapsid [7]. However, the molecular mechanisms underlying the modulation of HCV Core levels during HCV replication remain inadequately elucidated.

Reactive oxygen species (ROS) are highly active compounds formed from oxygen (O_2_). Even in normal cells, they are continuously generated during normal cellular activities, including protein folding within the endoplasmic reticulum, oxidative phosphorylation, and the metabolism of amino acids and lipids [8]. However, oxidative stress, leading to an imbalance between their elimination by scavenging molecules and the production of ROS, has been recognized as a significant contributor to hepatocyte death, inflammation, fibrogenesis, and the development of HCC [9]. ROS are generally regarded as detrimental byproducts of cellular metabolism because they can damage various biological molecules, including nucleic acids, proteins, and lipids [10]. H_2_O_2_ is particularly well-studied due to its crucial role in redox signaling, the modulation of oxidative stress, and its effect on various biological processes [11,12,13,14].

Previous studies have shown that ROS levels are frequently higher in the liver tissue of patients chronically infected with HCV, often by two to five orders of magnitude [15]. It has been further demonstrated that HCV infection in vitro can trigger the generation of ROS in cultured cells [1,16]. Several viral proteins, including Core, E1, E2, NS3, NS4B, and NS5A, have been implicated in inducing ROS production [17]. In particular, HCV Core is known to elevate mitochondrial ROS levels, primarily by reducing the activity of electron transport complex I or enhancing mitochondrial Ca^2+^ entry [18,19], contributing significantly to the oxidative stress in patients with chronic hepatitis C. The increased levels of ROS are believed to play a crucial role in the pathogenesis of HCV, especially in the progression towards HCC [20]. However, the precise role of ROS in the modulation of HCV replication remains uncertain, primarily because limited experimental evidence on this issue has been provided so far. Some studies have indicated that ROS suppresses HCV replication in human hepatoma cells [21,22], while others have postulated that ROS stimulate HCV replication by enhancing HCV protein synthesis [23,24]. Therefore, the precise effect of ROS on HCV replication is yet to be fully understood.

Previous studies have provided evidence demonstrating that the regulation of HCV Core levels is predominantly governed by an E3 ligase, E6-associated protein (E6AP) [25,26]. Additionally, it has been reported that the tumor suppressor p53 inhibits HCV replication by promoting the proteasomal degradation of HCV Core mediated by E6AP in cultured cells [27,28,29]. As it is relatively well established that p53 is promptly activated in response to DNA damage induced by ROS [30,31], it is interesting to explore whether ROS, such as H_2_O_2_, inhibit HCV replication by promoting the proteasomal degradation of HCV Core mediated by E6AP. To address this hypothesis, we initially examined whether H_2_O_2_ decreases HCV Core levels in a mechanism dependent on p53. Subsequently, we investigated whether E6AP is implicated in the p53-dependent proteasomal degradation of HCV Core induced by H_2_O_2_. We then investigated whether and how H_2_O_2_ activates *E6AP* expression in a p53-dependent manner. Finally, we attempted to demonstrate that H_2_O_2_ inhibits HCV replication by lowering HCV Core levels in an in vitro HCV replication system.

## 2. Materials and Methods

### 2.1. Plasmid

The plasmid pCMV-3 × HA1-Core [32] contains the complete HCV Core (genotype 1b) sequence positioned downstream of three copies of the influenza virus hemagglutinin (HA) epitope. p53 small hairpin RNA (shRNA), scrambled (SC) shRNA, and E6AP shRNA plasmids were obtained from Santa Cruz Biotechnology (Dallas, TX, USA). The plasmid pCMVT N-HA-hE6AP, encoding the complete human HA-tagged *E6AP*, was purchased from Addgene (Watertown, MA, USA). The pCMV p53-WT and pHA-ubiquitin (Ub) plasmids were generously provided by Dr. C.-W. Lee (Sungkyunkwan University, Korea) and Dr. Y. Xiong (University of North Carolina at Chapel Hill, NC, USA), respectively.

### 2.2. Cell Culture and Transfection

The HepG2 and Hep3B cell lines were provided by the Korean Cell Line Bank (KCLB). An immortalized hepatic cell line, HepaRG (Cat No. HPR101), was purchased from Biopredic International, Saint-Grégoire, France. Huh7D cells were kindly provided by Dr. S.M. Feinstone (US FDA). For transient expression, 4 × 10^5^ cells per 60 mm dish were transfected with designated plasmids utilizing TurboFect transfection reagent (Thermo Fisher Scientific, Waltham, MA, USA), according to the instructions of the manufacturer. All cells were cultured in Dulbecco Modified Eagle Medium (DMEM) (WelGENE, Gyeongsan, Republic of Korea) supplemented with 10% fetal bovine serum (Gibco, New York, NY, USA) and antibiotics (penicillin and streptomycin; Gibco). Cells were treated with H_2_O_2_, N-acetyl-L-cysteine (NAC), pifithrin-alpha (PFT-α), cycloheximide (CHX), MG132, or 5-aza-2′dC (Sigma, St. Louis, MO, USA), as needed.

### 2.3. In Vitro HCV Infection System

The plasmid pJFH-1, harboring the HCV cDNA derived from a Japanese patient with fulminant hepatitis downstream of a T7 promoter [33], was linearized at the 3′ end of the HCV cDNA through *Xba*I digestion. The linearized DNA was utilized as a template for in vitro transcription using MEGAscript (Thermo Fisher Scientific). Ten micrograms of JFH-1 RNA were introduced into Huh7D cells through electroporation, and viral stocks were subsequently prepared following previously established procedures [34]. Huh7D cells were subjected to either mock-infection or infection with HCV at a multiplicity of infection (MOI) of 10 for 24 h. Quantitative RT-PCR and conventional RT-PCR assays were conducted to detect HCV RNA levels, following previously described protocols [35].

### 2.4. Determination of Intracellular ROS Levels

Intracellular ROS levels were measured using chloromethyl dichlorodihydrofluorescein diacetate (CM-H_2_DCFDA; Invitrogen, Waltham, MA, USA), a widely applied probe specific for H_2_O_2_ in intact cells [36]. In brief, 1 × 10^5^ cells were treated with 10 µM CM-H_2_DCFDA for 30 min in serum-free media. After washing with PBS, cells were harvested through trypsin-EDTA treatment (Gibco). The conversion of CM-H_2_DCFDA into DCF, a green fluorescent product, was measured utilizing a microplate reader (Mithras LB940, Berthold Technologies, Bad Wildbad, Germany) with excitation and emission wavelengths set at 485 nm and 535 nm, respectively.

### 2.5. Western Blot Analysis

Cells were lysed in buffer supplemented with protease inhibitors (50 mM Tris-HCl, pH 7.5, 150 mM NaCl, 0.1% SDS, and 1% NP-40). Protein concentrations in cell lysates were quantified utilizing Protein Assay Dye Reagent (Bio-Rad, Hercules, CA, USA). Cell lysates were subjected to separation via SDS-polyacrylamide gel electrophoresis (SDS-PAGE) and transferred onto a nitrocellulose blotting membrane (GE Healthcare Life Science, Marlborough, MA, USA). Membranes were reacted with primary antibodies targeting HCV Core, DNA methyltransferase 1 (DNMT1) (Abcam, Cambridge, UK), Bax, DNMT3a, DNMT3b, HA, p53 upregulated modulator of apoptosis (PUMA), γ-tubulin, p21, p53 (Santa Cruz Biotechnology), and E6AP (Thermo Fisher Scientific). Subsequently, membranes were incubated with a horseradish peroxidase-conjugated secondary antibody, either anti-mouse or anti-rabbit IgG (H + L)-HRP (Bio-Rad). The protein bands were visualized using WesternBright ECL (Advansta, San Jose, CA, USA) and the ChemiDoc XRS imaging system (Bio-Rad).

### 2.6. Coomassie Brilliant Blue Staining

The cell lysates of each sample were separated by SDS-PAGE on an 8% acrylamide gel. Proteins on the gel were stained using Coomassie R-250 staining solution (0.05% Coomassie brilliant blue R-250 (Bio-Rad), 40% methanol, and 10% acetic acid) for 30 min. The gel was briefly rinsed with water and destained using a destaining solution (40% methanol and 10% acetic acid).

### 2.7. DNMT Activity Assay

DNMT activity in the nuclear extracts was quantified using an EpiQuik DNMT Activity/Inhibition Assay Ultra Kit (Epigentek, Farmingdale, NY, USA), following the instructions of the manufacturer. Briefly, nuclear extracts (10 μg) prepared from the cells were incubated with a universal DNMT substrate coated on microplate wells for 2 h at 37 °C. After adding a capture antibody (anti-5-methylcytosine antibody), a detection antibody, and then a color-developing solution in sequence, the amount of methylated DNA was measured by reading the optical density at a wavelength of 450 nm in a microplate reader (Bio-Rad).

### 2.8. Methylation-Specific PCR (MSP)

Genomic DNA was isolated from cells utilizing the QIAamp DNA Mini Kit (Qiagen, Hilden, Germany). Subsequently, bisulfite modification of the genomic DNA (1 µg) was performed utilizing an EpiTect Bisulfite kit (Qiagen), following the instructions of the manufacturer. The modified DNA (100 ng) was subjected to MSP for E6AP using a methylated primer pair, E6AP-Me-1F (5′-TTT TTA ATG GTT TGT GTG TC-3′) and E6AP-Me-1R (5′-TAC AAA CAA CGC ACA CCG-3′), and an unmethylated primer pair, E6AP-Un-1F (5′-TTT TTA ATG GTT TGT GTG TT-3′) and E6AP-Un-2R (5′-CAC ACA AAT CTC ACA ACC A-3′), as previously described [37].

### 2.9. Immunoprecipitation (IP)

An IP assay was conducted using a Classic Magnetic IP/Co-IP assay kit (Pierce), according to the specifications of the manufacturer. In brief, 4 × 10^5^ cells were transiently transfected with the specified plasmids for 48 h. Whole cell lysates (500 µg) were incubated with an anti-HCV Core antibody (Abnova, Taipei City, Taiwan) overnight at 4 °C to facilitate the formation of the immune complexes. Protein A/G magnetic beads (0.25 mg) were subsequently added, and the lysates were further incubated for an extra 1 h. The beads were collected using a magnetic stand (Pierce, Appleton, WI, USA), and the eluted antigen/antibody complexes were analyzed by Western blotting using the designated antibodies.

### 2.10. Cell Viability Analysis

For assessing cell viability, an MTT assay was conducted following previously described procedures [38]. In brief, cells were plated at a density of 1 × 10^4^ cells in 96-well plates and cultured under the specified conditions. The cells were subsequently exposed to 10 µM 3-(4,5-dimethylthiazol-2-yl)-2,5-diphenyltetrazolium bromide (MTT, Sigma-Aldrich) at 37 °C for 4 h. The formazan compounds generated from MTT by mitochondrial dehydrogenases of the viable cells were subsequently dissolved in DMSO (Sigma-Aldrich) and quantified by reading absorbance at 550 nm.

### 2.11. Statistical Methods

The values represent mean ± standard deviation (SD) from a minimum of three independent experiments. Two-tailed Student’s *t*-tests were employed for statistical analyses. A *p* value > 0.05 was deemed statistically nonsignificant, while a *p* value ≤ 0.05 was deemed statistically significant.

## 3. Results

### 3.1. H_2_O_2_ Inhibits HCV Replication in Human Hepatoma Cells

Initially, we examined whether H_2_O_2_ affects HCV replication in Huh7D cells, a human hepatoma cell line, which supports the efficient replication of HCV [39]. In line with a prior study [16], HCV increased intracellular ROS levels during its replication in Huh7D cells (Figure 1c). In addition, H_2_O_2_ treatment led to an additional dose-dependent increase in ROS levels. These results indicate that HCV infection and H_2_O_2_ treatment, either individually or in combination, result in an upregulation of intracellular ROS levels in human hepatoma cells. Interestingly, treatment with H_2_O_2_ decreased the levels of intracellular HCV Core and extracellular virus particles in Huh7D cells infected with HCV (Figure 1a,b). The inhibitory effect of H_2_O_2_ on HCV replication appears to be specific because significant adverse effects on MTT activity, an indicator of cell proliferation, cytotoxicity, and viability, or on the general pattern of protein synthesis in host cells, were not detected under our experimental conditions (Figure 1a,d).

To verify the inhibitory effect of H_2_O_2_ on HCV replication, we examined whether NAC, a representative antioxidant, could nullify the effect of H_2_O_2_ treatment on HCV replication in Huh7D cells. As expected, NAC treatment successfully prevented the increase in intracellular ROS levels in Huh7D induced by HCV infection and H_2_O_2_ treatment (Figure 1g). In addition, NAC treatment almost completely nullified the inhibitory effect of H_2_O_2_ on HCV replication in Huh7D cells, as evidenced by the increased levels of intracellular HCV Core and extracellular virus particles (Figure 1e,f), without affecting the MTT values in these cells (Figure 1h). NAC treatment also increased HCV replication in Huh7D cells, even in the absence of H_2_O_2_ treatment, possibly by removing the ROS induced by HCV itself (Figure 1e,f). Therefore, we conclude that H_2_O_2_ inhibits HCV replication in human hepatoma cells.

### 3.2. H_2_O_2_ Downregulates HCV Core Levels in a p53-Dependent Manner

As shown in Figure 1a, H_2_O_2_ dose-dependently downregulated the levels of HCV Core, a crucial determinant of HCV replication [40], during HCV replication in Huh7D cells. To confirm that the downregulation of HCV Core levels is a cause and not a consequence of HCV inhibition by H_2_O_2_, we examined whether H_2_O_2_ could directly reduce HCV Core levels without the involvement of other HCV proteins. Consistent with a previous report [20], ectopic *HCV Core* expression increased intracellular ROS levels in HepG2 but not Hep3B cells (Figure 2d,e). Additionally, H_2_O_2_ treatment in the presence of HCV Core resulted in an additional dose-dependent elevation of ROS levels in HepG2, while the effect was not significant in Hep3B cells (Figure 2d,e). H_2_O_2_ induced a dose-dependent reduction in the levels of ectopically expressed HCV Core in HepG2 (Figure 2a) but not Hep3B cells (Figure 2b), indicating the critical function of p53 in the H_2_O_2_-mediated downregulation of HCV Core. In HepG2 cells, treatment with H_2_O_2_ led to a decrease in MTT values, whereas the ectopic expression of *HCV Core* increased MTT values (Figure 2g), as previously demonstrated [41]. However, the inhibitory effect of H_2_O_2_ on cell viability was not observed in HepG2 cells expressing *HCV Core* (Figure 2g). This is probably because HCV Core restrained H_2_O_2_ from upregulating p53 and intracellular ROS levels in these cells (Figure 2a,d). Neither the negative effect of H_2_O_2_ nor the positive effect of HCV Core on MTT activity was observed in Hep3B cells (Figure 2h). These findings indicate that the downregulation of HCV Core levels induced by H_2_O_2_ is a specific event in the presence of p53.

We further explored whether the differential effects of H_2_O_2_ on HCV Core levels in HepG2 and Hep3B cells were attributed to differences in intracellular ROS levels between these two cell lines. Indeed, the ability of HCV Core and H_2_O_2_ to elevate intracellular ROS levels was more pronounced in HepG2 cells compared to Hep3B cells (Figure 2d,e). This difference in intracellular ROS levels when *HCV Core* is expressed and during H_2_O_2_ treatment can be attributed to the p53-mediated ROS amplification in HepG2 cells [42]. Interestingly, however, it was impossible to observe the downregulation of HCV Core levels and changes in MTT values in Hep3B cells treated with 800 μM H_2_O_2_ (Figure 2c,i), which raised intracellular ROS levels to the levels achieved in HepG2 cells treated with 200 μM H_2_O_2_ treatment (Figure 2f). This suggests that p53 may have an additional role, beyond amplifying ROS levels, in the downregulation of HCV Core levels by H_2_O_2_. Taken together, our findings indicate that H_2_O_2_ reduces HCV Core levels in a manner dependent on p53.

### 3.3. H_2_O_2_ Downregulates HCV Core Levels by Upregulating p53 Levels in Human Hepatoma Cells

Next, we explored the mechanism through which H_2_O_2_ reduces HCV Core levels in a p53-dependent manner. As reported previously [43], HCV infection increased p53 levels in Huh7D cells (Figure 1a). H_2_O_2_ treatment resulted in an additional dose-dependent increase in p53 levels in Huh7D cells. Moreover, the ectopic expression of *HCV Core* in HepG2 cells increased p53 levels (Figure 2a), as previously reported [43], and this effect was further enhanced by H_2_O_2_ treatment in a dose-dependent manner (Figure 2a). Notably, neither HCV infection nor H_2_O_2_ treatment could increase p53 levels in Huh7D cells in the presence of NAC (Figure 1e), indicating that the transcriptional activity of p53 is essential for its upregulation by *HCV Core* expression and H_2_O_2_ treatment. As a result, the intracellular ROS levels consistently correlated with p53 levels in the presence of HCV Core in both HCV infection (Figure 1a,c) and *HCV Core* overexpression systems (Figure 2a,d). Under the same conditions, a negative correlation between HCV Core and p53 levels was observed (Figure 1a and Figure 2a), consistently with the established function of p53 as an inhibitor of HCV replication [27,28,29]. These findings suggest that H_2_O_2_ increases p53 levels to downregulate HCV Core levels in human hepatoma cells.

To further confirm that H_2_O_2_ downregulates HCV Core levels by increasing p53 levels, we aimed to knock down *p53* in HepG2 cells expressing *HCV Core* in the presence of H_2_O_2_. The knockdown of *p53* dose-dependently nullified the ability of H_2_O_2_ to reduce HCV Core levels in HepG2 cells (Figure 3a). Furthermore, the ectopic expression of *p53* dose-dependently allowed H_2_O_2_ to lower HCV Core levels in Hep3B cells, in which ectopic p53 levels were also upregulated by H_2_O_2_ (Figure 3b). Notably, even without H_2_O_2_ treatment, the ectopic expression of *p53* alone was sufficient to reduce HCV Core levels in both HepG2 and Hep3B cells in a dose-dependent manner (Figure 3c,d). Taken together, we conclude that H_2_O_2_ decreases HCV Core levels by elevating p53 levels in human hepatoma cells.

To explore the necessity of p53′s transcriptional activity in the H_2_O_2_-mediated downregulation of HCV Core levels in human hepatoma cells, we employed a specific p53 inhibitor called PFT-α. This inhibitor is recognized for preventing the transactivation of *p53* responsive genes [44]. When treated with PFT-α, the ability of p53 to upregulate its target genes, including *p21*, *Bax*, and *PUMA*, was almost completely inhibited in both HepG2 cells and Hep3B cells expressing ectopic *p53*. This indicates that PFT-α effectively suppressed the transcriptional activity of p53 in these cells. Additionally, PFT-α decreased the levels of both endogenous p53 and exogenous p53 in HepG2 and Hep3B cells, respectively, regardless of the presence of HCV Core alone or in combination with H_2_O_2_ (Figure 3e,f). This downregulation is likely due to the inhibitory effect of PFT-α on the p53 amplification pathway, which relies on the p53 transcriptional activity [42]. Treatment with PFT-α upregulated HCV Core levels, notably in the presence of H_2_O_2_, in the same cells (Figure 3e,f). Interestingly, even in the presence of PFT-α, H_2_O_2_ retained its potential to decrease HCV Core levels in HepG2 and Hep3B cells. These findings suggest that p53 participates in the H_2_O_2_-mediated downregulation of HCV Core levels through at least two distinct mechanisms: one that relies on p53 transcriptional activity and another that operates independently of it.

### 3.4. H_2_O_2_ Increases E6AP Levels to Downregulate HCV Core Levels in a p53-Dependent Manner

As previously demonstrated [26], HCV Core was found to downregulate the levels of E6AP, an E3 ligase of HCV Core [25], in both HCV infection and *HCV Core* overexpression systems (Figure 1a and Figure 2a). Interestingly, H_2_O_2_ treatment led to an elevation of E6AP levels in both HCV-infected Huh7D cells (Figure 1a) and HepG2 cells expressing *HCV Core* (Figure 2a). The same outcome was also observed in Hep3B co-expressing both *HCV Core* and *p53*, but not in cells expressing *HCV Core* in the absence of p53 (Figure 2b and Figure 3b). Furthermore, ectopic *p53* overexpression in the presence of HCV Core without H_2_O_2_ treatment was enough to increase E6AP levels in HepG2 and Hep3B cells (Figure 3c,d). As a result, E6AP levels consistently correlated with p53 levels under these experimental conditions. These findings strongly indicate that, in the presence of HCV Core, H_2_O_2_ upregulates E6AP levels by increasing p53 levels in human hepatoma cells. Indeed, the potential of H_2_O_2_ to upregulate E6AP levels was almost completely nullified when p53 levels were reduced in HCV-infected Huh7D cells and *HCV Core*-expressing HepG2 cells, either by NAC treatment or *p53* knockdown, respectively (Figure 1e and Figure 3a).

The relationship between H_2_O_2_, p53, E6AP, and HCV Core levels in human hepatoma cells was further explored. In both HCV infection and *HCV Core* overexpression systems, E6AP levels were inversely proportional to HCV Core levels in the presence of p53 (Figure 1a and Figure 2a). However, this effect was not observed in Hep3B cells in the absence of p53 (Figure 2b). In addition, the ability of H_2_O_2_ to downregulate HCV Core levels was significantly diminished when E6AP levels were reduced, either through NAC treatment or *p53* knockdown (Figure 1e and Figure 3a). To further confirm the involvement of E6AP in the downregulation of HCV Core levels by H_2_O_2_, *E6AP* was knocked down in the presence of H_2_O_2_ in *HCV Core*-expressing HepG2 cells. The gradual downregulation of E6AP led to a stepwise reduction in the ability of H_2_O_2_ to downregulate HCV Core levels (Figure 4a). Moreover, the ectopic expression of *E6AP* in the absence of H_2_O_2_ was enough to decrease HCV Core levels in HepG2 cells (Figure 4b), as previously demonstrated [25]. These results indicate that p53 primarily acts in the H_2_O_2_-mediated downregulation of HCV Core levels by upregulating E6AP levels. Interestingly, however, the fact that ectopic *E6AP* expression in the absence of p53 failed to decrease HCV Core levels in Hep3B cells (Figure 4c) suggests that p53 may have an additional role beyond upregulating E6AP levels in the downregulation of HCV Core levels by H_2_O_2_.

### 3.5. H_2_O_2_ Stimulates E6AP Expression through Promoter Hypomethylation in the Presence of HCV Core

Next, we investigated how HCV Core and H_2_O_2_ individually or in combination regulate E6AP levels. In accordance with a prior report [26], HCV Core increased the levels of DNMT1, DNMT3a, and DNMT3b and their enzyme activity. This led to the promoter hypermethylation of the *E6AP* gene and subsequent downregulation of E6AP levels in HepG2 cells (Figure 5a,c). These effects were dependent on p53, as they were not observed in Hep3B cells (Figure 5b,d). Similarly, H_2_O_2_ induced the activation of DNMTs, leading to the inhibition of *E6AP* expression through promoter hypomethylation in HepG2 (Figure 5a,c) but not in Hep3B cells (Figure 5b,d, lane 3). Interestingly, when H_2_O_2_ was combined with HCV Core in HepG2 cells, it downregulated both the enzyme activities and protein levels of DNMT1, DNMT3a, and DNMT3b, leading to promoter hypomethylation of the *E6AP* gene. Consequently, this resulted in the upregulation of E6AP protein levels in HepG2 cells (Figure 5a,c). These effects were not observed in Hep3B cells (Figure 5b,d). The individual and combined effects of HCV Core and H_2_O_2_ on E6AP and HCV Core levels in HepG2 cells were almost completely nullified by treatment with a universal DNMT inhibitor, 5-Aza-2′dC (Figure 5e). Taken together, we conclude that H_2_O_2_ activates *E6AP* expression via promoter hypomethylation, which is dependent on both p53 and HCV Core.

### 3.6. H_2_O_2_ Downregulates HCV Core Levels through the Promotion of Proteasomal Degradation Mediated by E6AP in a p53-Dependent Mechanism

A previous report has demonstrated that E6AP as an E3 ligase induces ubiquitination and proteasomal degradation of HCV Core [25]. Therefore, we first determined whether H_2_O_2_ reduces the protein stability of HCV Core in human hepatoma cells. For this purpose, HepG2 cells expressing *HCV Core* were treated with CHX to inhibit additional protein synthesis and we measured the levels of both HCV Core and γ-tubulin in these cells. The half-life (t_1/2_) of HCV Core was 65.34 min, which was shortened to 26.02 min upon treatment with H_2_O_2_ in HepG2 cells (Figure 6a), suggesting that H_2_O_2_ decreases the protein stability of HCV Core in HepG2 cells. However, the t_1/2_ value of HCV Core in Hep3B cells was 124.59 min, showing minimal impact from H_2_O_2_ treatment (t_1/2_ = 141.75 min). These results suggest a p53-dependent mechanism by which H_2_O_2_ reduces the protein stability of HCV Core.

Having established that H_2_O_2_ shortens the half-life of HCV Core, we further examined whether H_2_O_2_ would enhance the ubiquitination of HCV Core protein by upregulating E6AP levels. For this purpose, we transfected HepG2 cells with HA-tagged Ub along with HCV Core, both in the presence and absence of H_2_O_2_ treatment, and immunoprecipitated the Ub-complexed HCV Core. The data from the co-IP clearly showed that E6AP interacted with HCV Core, resulting in its ubiquitination as demonstrated by the detection of multiple smeared bands corresponding to Ub(n)-HCV Core (Figure 6b, lane 2). Interestingly, H_2_O_2_ increased the interaction between the HCV Core and E6AP, leading to robust ubiquitination of HCV Core and a subsequent reduction in protein levels (Figure 6b, lane 3). Furthermore, *E6AP* knockdown in the H_2_O_2_-treated *HCV Core*-expressing HepG2 cells decreased the interaction between HCV Core and E6AP, leading to a reduction in the ubiquitination of HCV Core and an increase in its protein levels (Figure 6b, lane 4). This supports the conclusion that H_2_O_2_ increases the ubiquitination and proteasomal degradation of HCV Core by upregulating E6AP levels in HepG2 cells. Although E6AP could interact with HCV Core to induce its ubiquitination in Hep3B cells (Figure 6b, lane 6), the impact of H_2_O_2_ on the interaction between HCV Core and E6AP, as well as the subsequent ubiquitination of HCV Core, was largely negligible in Hep3B cells (Figure 6b, lane 7). Treatment with MG132 nearly nullified the ability of H_2_O_2_ to downregulate HCV Core levels and equalized the protein levels of HCV Core in the absence or presence of H_2_O_2_ in HepG2 cells (Figure 6c), confirming that H_2_O_2_ downregulates HCV Core levels by inducing proteasomal degradation mediated by E6AP. Based on these observations, we conclude that H_2_O_2_ induces the E6AP-mediated proteasomal degradation of HCV Core by upregulating E6AP levels in a p53-dependent manner.

### 3.7. H_2_O_2_ Inhibits HCV Replication by Stimulating E6AP-Mediated Ubiquitination and Proteasomal Degradation of HCV Core

Consistent with the observations in the *HCV Core* overexpression system (Figure 2), HCV infection led to an increase in p53 levels but a decrease in E6AP levels, likely attributed to the influence of HCV Core, during HCV replication in both Huh7D and HepaRG cells (Figure 7a,c). Treatment with H_2_O_2_ resulted in an additional upregulation of both p53 and E6AP levels, consequently inhibiting HCV replication in both Huh7D and HepaRG cells, as evidenced by reduced levels of intracellular HCV Core (Figure 7a,c) and extracellular HCV particles (Figure 7b,d). Additionally, *E6AP* knockdown nearly nullified the capability of H_2_O_2_ to inhibit HCV replication in Huh7D cells, which was evidenced by an increase in intracellular HCV Core levels as well as extracellular HCV particles (Figure 7a,b). Treatment with H_2_O_2_ also enhanced the interaction between HCV Core and E6AP during HCV infection in Huh7D cells, leading to robust ubiquitination of HCV Core and the subsequent reduction in its protein levels (Figure 7C). These effects were almost nullified when *E6AP* was knocked down by a specific shRNA (Figure 7C, lane 4). Taken together, we conclude that H_2_O_2_ inhibits HCV replication in vitro by upregulating E6AP levels and promoting the Ub-dependent proteasomal degradation of HCV Core.

## 4. Discussion

The effects of ROS on viruses are diverse and can vary depending on the specific virus, the host cell type, and the stage of the viral life cycle. For example, ROS generated in host cells may exhibit antiviral effects against hepatitis B virus (HBV) [45]. However, HBV independently develops antioxidant defense mechanisms, enhancing resistance to ROS [46]. It is important to note that the relationship between ROS and viruses is intricate, and the specific effects can vary widely among different viruses. The present study demonstrated that ROS like H_2_O_2_ inhibit HCV replication and this effect is dependent on the presence of p53.

HCV infection is closely associated with oxidative stress, marked by increased levels of ROS in the liver and blood of infected individuals [15,47]. This process involves both viral and host factors. In the context of host factors, the activation of virus-specific cytotoxic T lymphocytes (CTLs) is a fundamental mechanism in eliminating viruses during infection [48]. CTLs primarily target infected hepatocytes, leading to their destruction and triggering the production of inflammatory cytokines [49]. This immune response substantially contributes to generating ROS within the liver. On the viral side, HCV itself, particularly HCV Core, contributes to the accumulation of ROS within hepatocytes undergoing infection [1,18]. The present study confirms that both HCV infection and *HCV Core* expression lead to higher levels of ROS in human hepatoma cells (Figure 1 and Figure 2), but their effects were still limited compared to what would occur during an actual infection with an intact immune response [15,47]. To more accurately evaluate the effect of ROS on HCV replication in these cultured cells, the present study used H_2_O_2_ to induce controlled and sufficient levels of intracellular ROS.

The relationship between ROS and HCV replication is rather complex, and different studies have reported conflicting results. Early study has suggested that increased ROS levels impair viral replication. For example, ROS, particularly H_2_O_2_, can suppress HCV RNA replication in human hepatoma cells. A more recent study also shown the sensitivity of HCV proteins, such as HCV Core and NS5A, to degradation induced by oxidative stress, specifically the superoxide anion donor menadione [22]. Consistent with the adverse impact of oxidative stress on HCV in vitro, certain antioxidants, including resveratrol, vitamins E, and vitamin A, have been demonstrated to stimulate HCV replication, with mechanisms that remain undisclosed [50]. In contrast, there are studies suggesting that H_2_O_2_ can enhance HCV replication. For instance, H_2_O_2_ can modify factors involved in the HCV internal ribosome entry site (IRES)-dependent translation of HCV RNA, potentially promoting viral replication [23]. Additionally, the removal of ROS through antioxidant treatment has been shown to inhibit HCV replication [24]. The current study provides evidence that H_2_O_2_ can inhibit HCV replication in cultured human hepatoma cells (Figure 1a,b). The finding that the antioxidant NAC can prevent the inhibitory effect of H_2_O_2_ further supports the conclusion that H_2_O_2_ has an inhibitory effect on HCV replication (Figure 1e,f). Essentially, the effect of H_2_O_2_ on HCV replication appears to be context-dependent and may vary depending on the specific stages of the viral life cycle. Understanding these complexities is essential for the development of strategies to effectively control HCV infection and related liver diseases. H_2_O_2_ is known to induce single- and double-stranded DNA breaks, which trigger the activation of DNA damage signal transduction pathways and subsequently lead to the upregulation of p53 levels [14]. The present study showed that H_2_O_2_ downregulates HCV Core levels in HepG2 cells, where H_2_O_2_ upregulated p53 levels (Figure 2a). However, this effect was not observed in Hep3B cells, where p53 was absent (Figure 1b). Ectopic *p53* expression restored the potential of H_2_O_2_ to downregulate HCV Core levels in Hep3B cells, in which H_2_O_2_ upregulated exogenous p53 levels (Figure 3b). Additionally, *p53* knockdown nearly nullified the ability of H_2_O_2_ to decrease HCV Core levels in HepG2 cells (Figure 3a). Therefore, it is evident that p53 plays a crucial role in the regulation of HCV Core levels and the subsequent inhibition of HCV replication by H_2_O_2_. This finding aligns with the function of p53 as a suppressor of HCV replication [27,28,29].

According to the present study, H_2_O_2_ treatment led to the elevation of intracellular ROS levels, particularly greater in HepG2 compared to Hep3B cells (Figure 2d,e). This discrepancy in ROS levels can be explained by the p53-mediated increase in ROS levels [42]. While it might be tempting to assume that the p53 plays a role in the H_2_O_2_-induced downregulation of HCV Core levels by increasing intracellular ROS levels in HepG2 cells, it is intriguing that even when similar levels of intracellular ROS were achieved in both HepG2 and Hep3B cells through H_2_O_2_ treatment at different concentrations (200 µM for HepG2 and 800 µM for Hep3B), the negative effect of H_2_O_2_ on HCV Core levels was only observed in HepG2 cells (Figure 2). Hence, more intricate functions of p53 appear to participate in the H_2_O_2_-induced inhibition of HCV replication.

A previous study has shown that E6AP as an E3 ligase of HCV Core mediates the ability of p53 to decrease HCV Core levels [27]. This report also demonstrated that H_2_O_2_ upregulated E6AP levels in both *HCV Core* overexpression and HCV replication systems by upregulating p53 levels (Figure 1, Figure 2 and Figure 3). Furthermore, the present study provided evidence that H_2_O_2_ inhibits HCV replication by reducing HCV Core levels through E6AP-mediated proteasomal degradation. This evidence includes H_2_O_2_ inhibiting HCV replication by increasing E6AP levels and decreasing HCV Core levels (Figure 1), H_2_O_2_ inducing the E6AP-mediated polyubiquitination and proteasomal degradation of HCV Core in a p53-dependent manner (Figure 6), and all these effects becoming ineffective when *E6AP* was knocked down (Figure 4, Figure 6 and Figure 7), substantiating the direct involvement of E6AP in the H_2_O_2_-induced reduction of HCV Core levels and the subsequent inhibition of HCV replication.

*E6AP* expression is predominantly regulated through DNA methylation, especially in the presence of HCV Core [26]. In this context, both HCV Core and H_2_O_2_ independently increased the levels of DNMT1, DNMT3a, and DNMT3b as well as their enzyme activity, which led to the methylation of the CpG island in the *E6AP* gene promoter and consequently downregulated *E6AP* expression, but notably, this effect was dependent on the presence of p53 (Figure 5). Furthermore, p53 inhibits the expression of the DNMTs [51]. Interestingly, when H_2_O_2_ was combined with HCV Core in the presence of p53, an opposite effect was observed regarding the DNA methylation-mediated regulation of *E6AP* gene expression (Figure 5). The reasons behind these differential effects of HCV Core and H_2_O_2_ on the host DNA methylation system, depending on whether they act individually or in combination, remain unclear. The p53 levels may play a role in these differences. Another hypothesis is that HCV Core and H_2_O_2_ may regulate the host DNA methylation system through mechanisms not dependent on p53 levels. Further research is necessary to fully understand the underlying mechanisms.

The present study showed that both HCV infection and *HCV Core* expression result in elevated levels of p53 and intracellular ROS in human hepatoma cells (Figure 1 and Figure 2), as previously demonstrated [43]. The observation that HCV Core modulates its levels during HCV replication by creating an inhibitory feedback loop between ROS and p53 raises intriguing questions. While the exact reasons remain to be fully elucidated, there are several plausible explanations for this phenomenon. HCV Core might regulate its protein levels as a survival strategy during long-term chronic persistent infection. By maintaining its replication within a specific range, the virus could avoid triggering an overwhelming host immune response, thereby increasing its chances of persisting in the host. On the other hand, the suppression of HCV replication by H_2_O_2_ and the subsequent decrease in HCV Core levels could act as a natural host defense mechanism against HCV infection. In this scenario, the host defense system, particularly through the action of p53 and the regulation of ROS, plays a role in controlling and limiting HCV replication. The function of p53 as a guardian of the host genome by controlling HCV replication through ROS regulation adds another layer of complexity to the host–virus interaction. In this respect, human primary hepatocytes and in vivo animal experiments, particularly those involving humanized mice, can serve as valuable tools for a thorough examination of the impact of H_2_O_2_ on HCV replication. By using human primary hepatocytes, we can study the effects of oxidative stress on HCV in a more biologically relevant environment, simulating conditions closer to those in the human liver. Additionally, in vivo humanized mice may allow for a comprehensive understanding of the interplay between H_2_O_2_ and HCV within a living organism, providing insights that may be crucial for therapeutic interventions or furthering our understanding of viral pathogenesis. Further research is necessary to clarify these explanations and gain a more comprehensive understanding of how HCV Core, ROS, and p53 collectively influence HCV replication.

## 5. Conclusions

ROS, such as H_2_O_2_, can have an inhibitory effect on the replication of HCV in cultured human hepatoma cells. More specifically, H_2_O_2_ increases the levels of an E3 ligase called E6AP and triggers a process involving the ubiquitination and proteasomal degradation of HCV Core. Importantly, these effects are mediated by p53, a well-known tumor suppressor protein. These findings align with the established function of p53 as a suppressor of HCV replication. Therefore, this research provides valuable insights into the mechanisms through which p53 influences HCV replication, particularly in situations involving oxidative stress.

## Figures and Tables

**Figure 1 cells-13-00062-f001:**
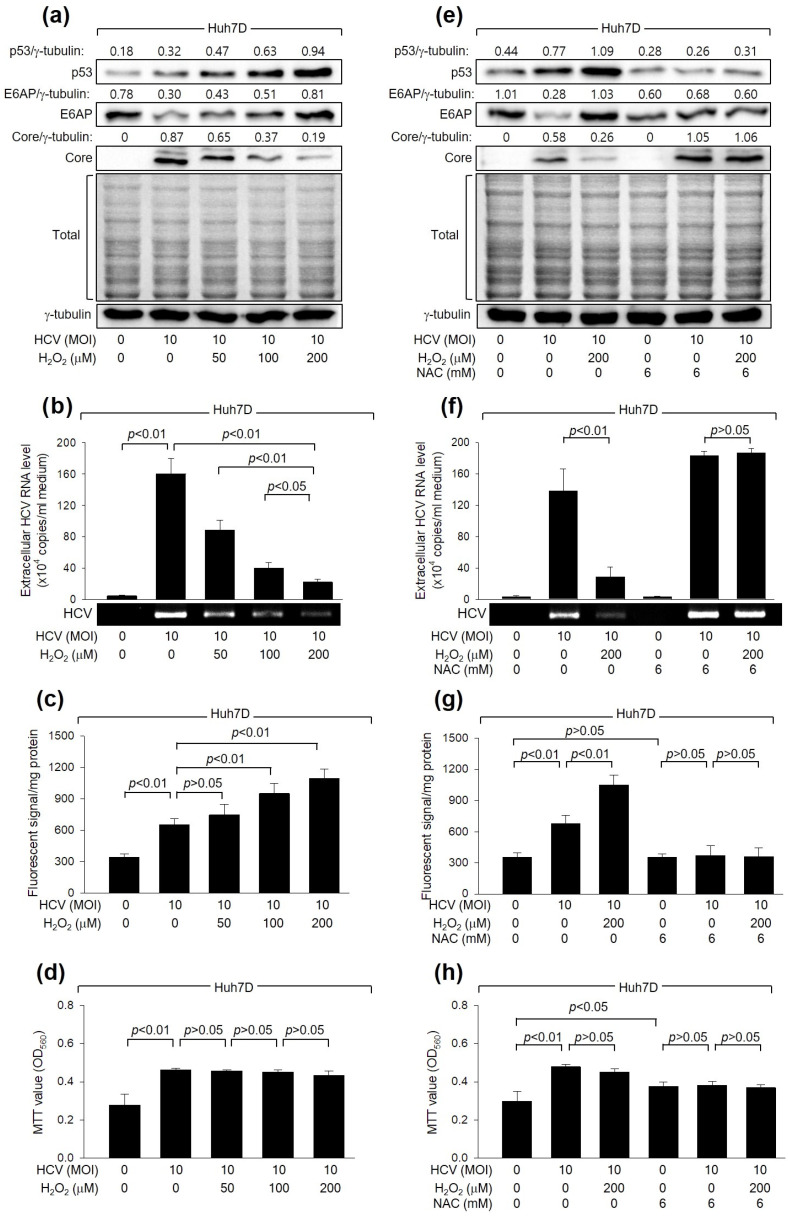
H_2_O_2_ inhibits HCV replication in human hepatoma cells. Huh7D cells infected with HCV for 24 h were subjected to treatment with H_2_O_2_ for an additional 24 h. Cells were treated with N-acetylcysteine (NAC), an antioxidant, during HCV replication (**e**–**h**). (**a**,**e**) Cell lysates were analyzed by Western blotting to detect the levels of the indicated proteins. The gels, stained with Coomassie solution, illustrate the total protein levels in each sample. (**b**,**f**) The levels of HCV particles in the culture supernatants, as prepared in (**a**,**e**), were quantified using both quantitative real-time RT-PCR (q-RT-PCR) (*n* = 4) and conventional RT-PCR. Results are shown as mean ± standard deviation from four independent experiments (*n* = 4). (**c**,**g**) Intracellular ROS levels were determined utilizing chloromethyl dichlorodihydrofluorescein diacetate (CM-H2DCFDA) (*n* = 4). (**d**,**h**) Cell viability was assessed by performing MTT assays (*n* = 4).

**Figure 2 cells-13-00062-f002:**
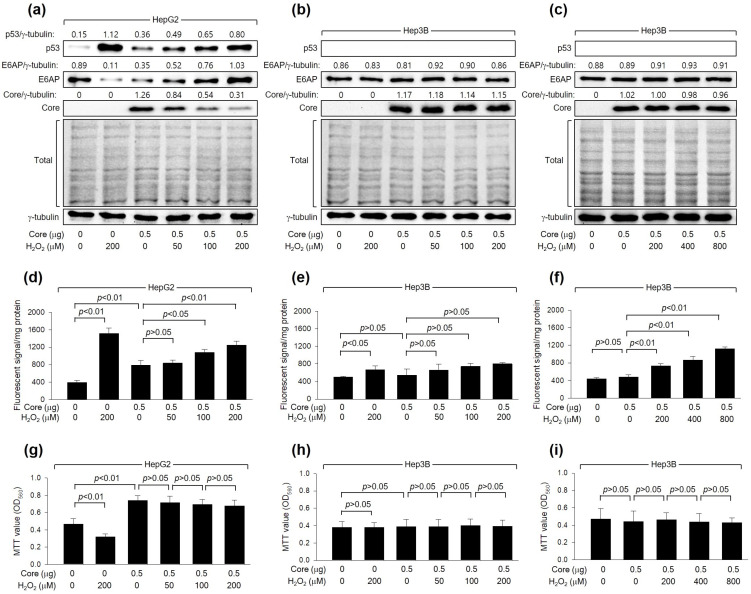
H_2_O_2_ downregulates HCV Core levels in a p53-dependent mechanism. Cells were transfected with the specified quantity of *HCV Core* expression plasmid for 24 h and subsequently treated with increasing concentrations of H_2_O_2_ for an extra 24 h. (**a**–**c**) Levels of the indicated proteins were determined by Western blotting. The SDS-PAGE gels were shown after staining with Coomassie brilliant blue. (**d**–**f**) Intracellular levels of ROS were measured as in Figure 1c (*n* = 3). (**g**–**i**) Cell viability was determined using MTT assays (*n* = 4).

**Figure 3 cells-13-00062-f003:**
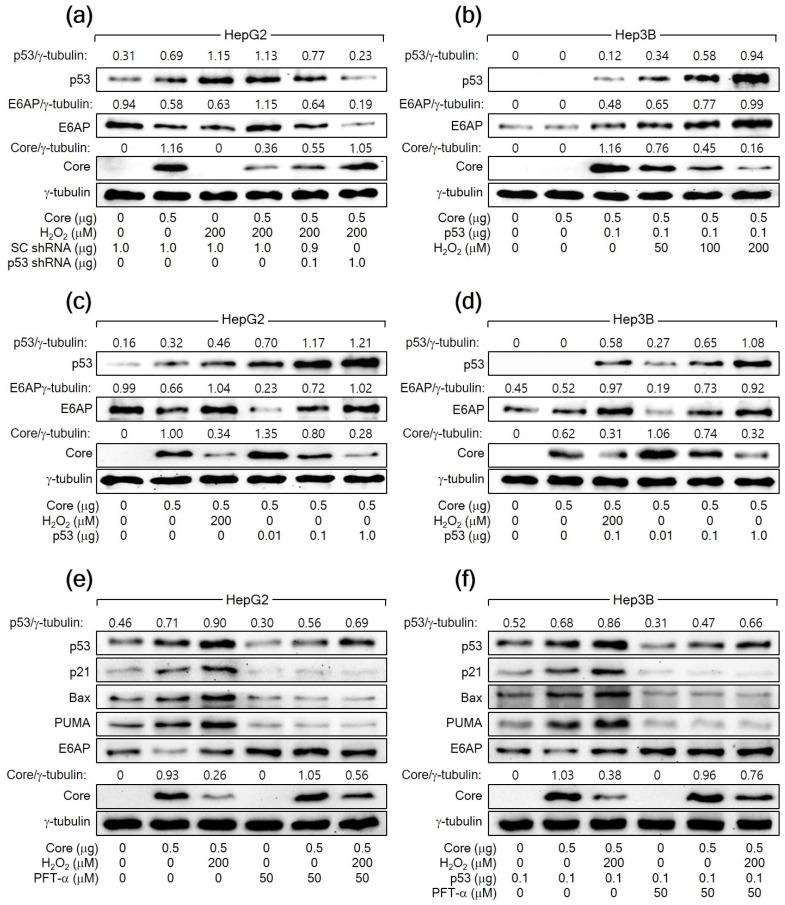
H_2_O_2_ downregulates HCV Core levels by elevating p53 levels in human hepatoma cells. (**a**–**d**) Cells were transiently transfected with the specified plasmids for 24 h and treated with H_2_O_2_ at the designated concentrations for an extra 24 h, analyzed by Western blotting. For (**e**,**f**), some cells were treated with H_2_O_2_ in the presence of pifithrin-α (PFT-α).

**Figure 4 cells-13-00062-f004:**
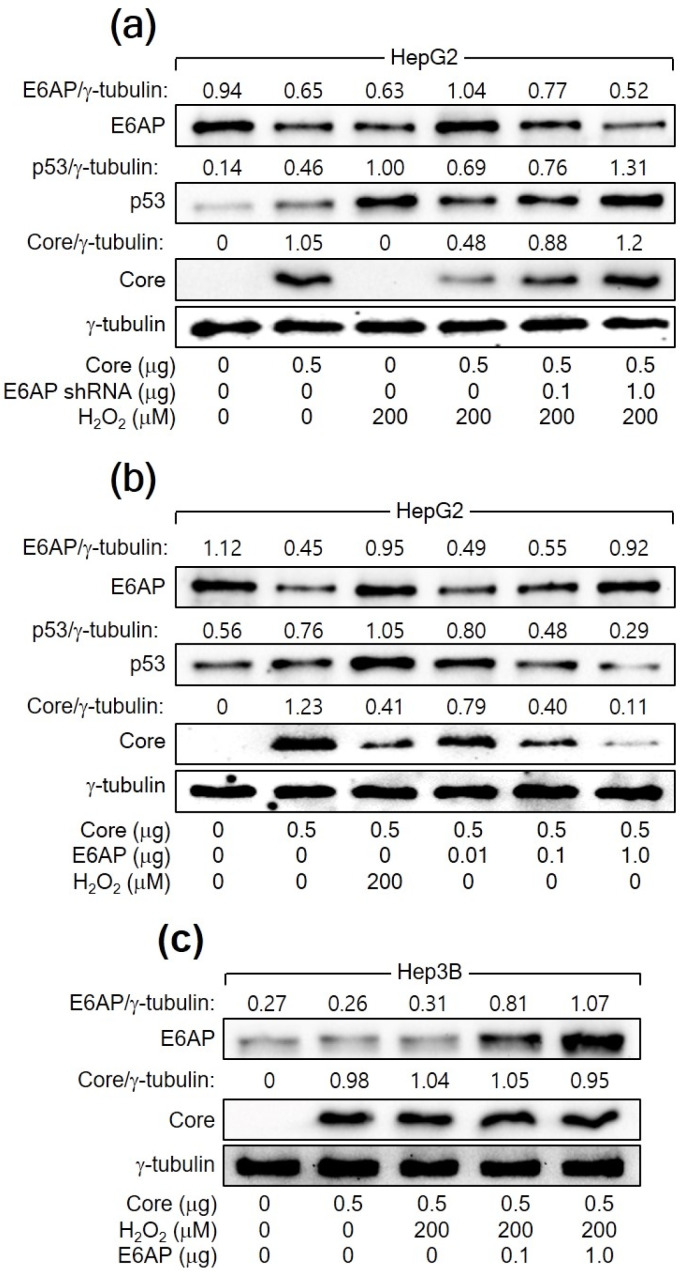
H_2_O_2_ downregulates HCV Core levels by upregulating E6AP levels in a p53-dependent manner. (**a**–**c**) Cells were transfected with the specified plasmids for 24 h and treated with H_2_O_2_ for an extra 24 h, followed by Western blotting.

**Figure 5 cells-13-00062-f005:**
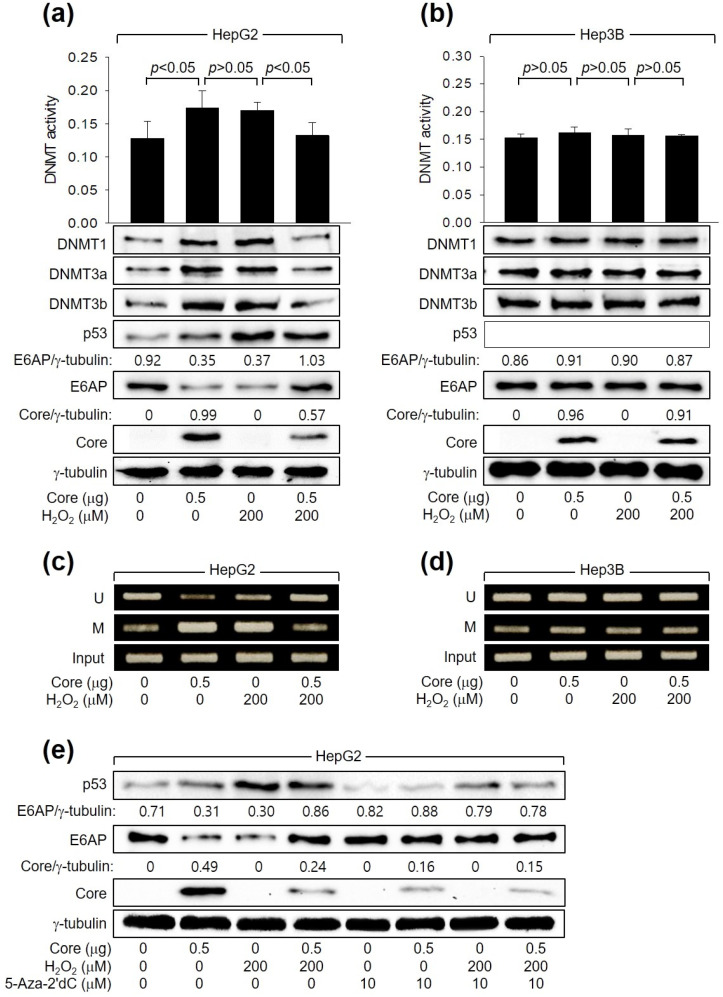
H_2_O_2_ stimulates *E6AP* expression through promoter hypomethylation in the presence of p53 and HCV Core. Cells were transfected with the specified plasmids for 24 h and subsequently treated with H_2_O_2_ for an extra 24 h. (**a**,**b**) DNMT activity from cells was determined (*n* = 3). Levels of the indicated proteins were measured by Western blotting. (**c**,**d**) Methylation-specific PCR (MSP) was performed to determine whether the CpG sites in the *E6AP* promoter are unmethylated (U) or methylated (M). (**e**) Cells were treated with the specified concentration of 5-Aza-2′dC for 24 h before harvesting.

**Figure 6 cells-13-00062-f006:**
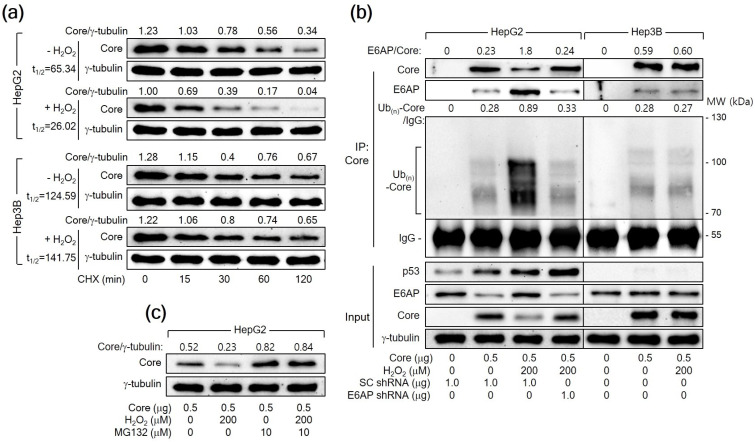
H_2_O_2_ triggers the E6AP-mediated ubiquitination and proteasomal degradation of HCV Core in a p53-dependent manner. (**a**) Cells prepared as in Figure 5a,b were subjected to treatment with 50 μM cycloheximide (CHX) for the specified duration before harvesting. The quantification of each band was performed using Image J image-analysis software (NIH, USA) to determine the half-life (t_1/2_) of HCV Core. The presented values represent the levels of HCV Core relative to the loading control (γ-tubulin). (**b**) Cells were transfected with the specified plasmids for 24 h and treated with H_2_O_2_ for an extra 24 h. The transfection mixtures included the *HA-Ub* expression plasmid. Total HCV Core proteins in cell lysates were immunoprecipitated using an anti-HCV Core antibody and subsequently analyzed by Western blotting. The membranes were probed with antibodies against p53, HCV Core, E6AP, and HA to detect p53, HCV Core, E6AP, and HA-Ub-complexed HCV Core, respectively. Additionally, the input shows the levels of the specified proteins in the cell lysates. (**c**) Cells were either mock-treated or treated with MG132 for 4 h before harvesting.

**Figure 7 cells-13-00062-f007:**
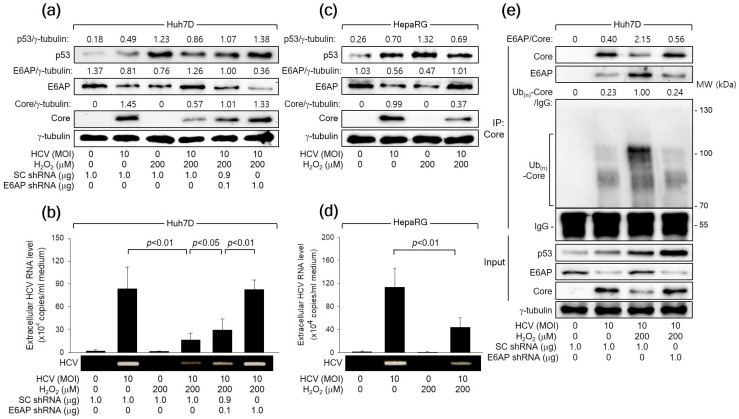
H_2_O_2_ inhibits HCV replication in vitro through the E6AP-mediated downregulation of HCV Core levels. (**a**) Cells were transfected with the designated plasmids for 24 h and infected with HCV for an extra 24 h in the absence or presence of H_2_O_2_, followed by Western blot analysis. (**b**) The levels of HCV particles in the supernatants, as prepared in (**a**), were determined using both q-RT-PCR (*n* = 4) and conventional RT-PCR. (**c**) Cells cultured in a differentiation medium (Biopredic International) for 2 weeks were infected with HCV for 24 h and treated with H_2_O_2_ for an extra 24 h, followed by Western blotting. (**d**) The levels of HCV particles in the supernatants prepared in (**c**) were determined using q-RT-PCR (*n* = 3) and conventional RT-PCR. (**e**) Cells were transfected with the specified plasmids for 24 h and subsequently infected with HCV for an extra 24 h, either in the absence or presence of H_2_O_2_, followed by co-IP, as described in Figure 6b.

## Data Availability

The data presented in this study are available from the corresponding author upon reasonable request.

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
