# Peer review of "Hydrogen Peroxide Inhibits Hepatitis C Virus Replication by Downregulating Hepatitis C Virus Core Levels through E6-Associated Protein-Mediated Proteasomal Degradation"

_cells, 2023, doi:10.3390/cells13010062_

Round 1

Reviewer 1 Report

Comments and Suggestions for Authors

The paper is well organized and the experiments are well thought out and executed.

The issue of the relationship of HCV and HCV replication to mitochondria is not entirely clear. Hepatitis C virus (HCV) infection can regulate the number and dynamics of mitochondria, and is associated with a prominent hepatic mitochondrial injury. Mitochondrial distress conveys oxidative damage, which is implicated in liver disease progression.

The authors have indicated in the introduction that there is some contradicting data about the role of ROS in the regulation of core. In addition, there are also studies indicating that the interactions of HCV proteins with mitochondrial and ER membranes (VDAC1 in particular) in fact do not affect membrane-associated functions in glucose homeostasis and Ca2+ signaling. This adds another controversy to the introduction to the present study.

What needs some improvement is the introduction into the problem and the discussion of the results. Example of what is lacking is information on other agents of oxidative stress and the effect that ROS have on infectious agents other than HCV. There needs to be a bit better distinction between the involvement of host and viral factors.

Lines 522-536 of the discussion are very appropriate. A sentence on a specific target project that can evolve form this work should be added.

Author Response

The comments on

1. Example of what is lacking is information on other agents of oxidative stress.

Answer: The information on other agents of oxidative stress has been added in the Discussion section on page 21, line 522-528 of the revised manuscript.

2. The effect that ROS have on infectious agents other than HCV.

Answer: The effect that ROS have on infectious agents other than HCV has been added to the Discussion section on page 20-21, line 494-502 of the revised manuscript.

3. There needs to be a bit better distinction between the involvement of host and viral factors. Lines 522-536 of the discussion are very appropriate. A sentence on a specific target project that can evolve form this work should be added.

Answer: The specific target project related to the distinction between the involvement of host and viral factors has been added to the Discussion section on page 23, line 607-615 of the revised manuscript.

Reviewer 2 Report

Comments and Suggestions for Authors

In this study, Yoon and Jang investigated the in vitro impact of oxidative stress on HCV replication. They obtained evidence that H2O2 inhibits HCV replication in human hepatoma cells, downregulates HCV Core levels in a p53-dependent manner, downregulates HCV Core levels by upregulating p53 levels in human hepatoma cells, increases E6AP levels to downregulate HCV Core levels in a p53-dependent manner, activates E6AP expression through promoter hypomethylation in the presence of HCV Core, downregulates HCV Core levels through the promotion of E6AP-mediated proteasomal degradation in a p53-dependent mechanism, and inhibits HCV replication by stimulating E6AP-mediated ubiquitination and proteasomal degradation of HCV Core. Based on these results, the authors concluded that oxidative stress inhibits HCV replication by decreasing Core levels via E6AP-mediated proteasomal degradation.

The experiments are straightforward, and the data well presented.

Major points:

1/ The old MTT cytotoxic assay is not sensitive compared to more recent fluorescent and luminescent assays for detecting viable cell numbers (i.e., CellTiter-Glo® 2.0 Cell Viability Assay). This is critical to repeat some of the important experiments with these assays to ensure that cell viability does not interfere with the interpretation of the results.

2/ The Western blot protein bands (gel) need to be quantified by a densitometer.

3/ The major experiments should be repeated with more relevant cells such as primary hepatocytes in order to eliminate the possibility that the phenotype is also observed in Huh7 cells.

Author Response

1. The old MTT cytotoxic assay is not sensitive compared to more recent fluorescent and luminescent assays for detecting viable cell numbers (i.e., CellTiter-Glo® 2.0 Cell Viability Assay). This is critical to repeat some of the important experiments with these assays to ensure that cell viability does not interfere with the interpretation of the results.

Answer: As the reviewer pointed out, the CellTiter-Glo® 2.0 cell viability assay may be a more sensitive and accurate method than the traditional MTT assay. However, the MTT assay we used in the present study is also a well-established and reliable method for assessing cell viability, and thus has extensively used in our previous studies.

2. The Western blot protein bands (gel) need to be quantified by a densitometer.

Answer: Most of the major western blot protein bands (gels) were quantified using Image J software (NIH) and presented in revised figures.

3. The major experiments should be repeated with more relevant cells such as primary hepatocytes in order to eliminate the possibility that the phenotype is also observed in Huh7 cells.

Answer: We performed some essential experiments using HepaRG cells, as shown in Figure 7c. As you known, HepaRG is an immortalized cell line that exhibit several physiological activities equivalent to human primary hepatocytes. In addition, we have a plan to do some critical experiments using primary hepatocytes and humanized mouse models as further studies, as summarized on page 23, lines 607-615.

Round 2

Reviewer 2 Report

Comments and Suggestions for Authors

Although the MTT assay is no longer used by "modern" research labs, the authors have addressed all previous issues.